# *Miyeokgui* (*Undaria pinnatifida* Sporophyll) Characteristic under Different Relative Humidity: Microbial Safety, Antioxidant Activity, Ascorbic Acid, Fucoxanthin, *α*-/*β*-/*γ*-Tocopherol Contents

**DOI:** 10.3390/foods12122342

**Published:** 2023-06-11

**Authors:** Kyo-Yeon Lee, Jong Min Kim, Jiyeon Chun, Ho Jin Heo, Chae Eun Park, Sung-Gil Choi

**Affiliations:** 1Department of Food Science and Technology, Institute of Agriculture and Life Science, Gyeongsang National University, Jinju 52828, Republic of Korea; leeyeon0511@naver.com (K.-Y.L.); myrock201@gnu.ac.kr (J.M.K.); hjher@gnu.ac.kr (H.J.H.); 2Division of Applied Life Science (BK21), Gyeongsang National University, Jinju 52828, Republic of Korea; 3Division of Food Science and Technology, Sunchon National University, Suncheon 57922, Republic of Korea; cjy-fall@scnu.ac.kr (J.C.);

**Keywords:** *Undaria pinnatifida* sporophyll, relative humidity, antioxidant, fucoxanthin, tocopherol

## Abstract

This study aimed to investigate the effects of different relative humidities (%) on the microbial safety, antioxidant activity, ascorbic acid, fucoxanthin, and tocopherol contents of *Undaria pinnatifida* sporophyll powder (UPSP) stored for 4 weeks. The caking phenomenon did not occur in the 11–53% relative humidity conditions, but it did in the 69%, 81%, and 93% relative humidity conditions with caking index values of 88.30%, 99.75%, and 99.98%, respectively. The aerobic bacterial contents increased drastically in samples stored at 69–93% relative humidity. Ascorbic acid was unstable at high relative humidity, but fucoxanthin and tocopherol were more unstable at low relative humidity. Therefore, it was most stable at intermediate relative humidity. The 69% relative humidity sample had higher DPPH (12.57 g BHAE/kg), ABTS (4.87 g AAE/kg), and FRAP (4.60 g Fe (II)/kg) than the other samples. This study could be helpful for the storage and transport of UPSP under optimum relative humidity conditions, which can significantly prevent quality losses.

## 1. Introduction

Seaweed consumption as a medicinal and food ingredient has been long established in East Asia, in countries such as Korea, China, and Japan [1,2]. Recently, the more vital awareness of health-related benefits associated with consuming seaweed-added products and materials has influenced customers’ choices. Thus, the international demand for seaweeds has been proliferating [3].

*Undaria pinnatifida* is a brown marine alga containing beneficial bioactive compounds. The body of *Undaria pinnatifida* is made up of a blade (lamina), midrib, sporophyll, and root-like structures [4]. Functional materials include polysaccharides (fucoidan), carotenoids (fucoxanthin), peptides, polyphenols, antioxidants, polyunsaturated fatty acids, dietary fibers, minerals, vitamins, and phytochemicals [5]. Fucoxanthin is a representative bioactive ingredient of UPS, one of the most abundant natural carotenoids, and is found primarily in marine macro and microalgae [6,7]. Fucoxanthin, an allenic carotenoid, is also renowned for its various properties, such as antioxidant, anticancer, antiobesity, and antidiabetic properties [8,9,10]. In addition, the main vitamins in brown seaweeds are vitamins A to C and E [11]. This seaweed is high in nutrients and natural bioactive chemicals, making it a suitable resource for functional food and pharmaceutical sectors [12].

Harvested seaweed contains a significant amount of water. Seaweed, containing 75% to 85% water while fresh, with the first signs of quality loss, including silting, color change, and the production of an off-odor and dirt on the leaves [13], is frequently distributed as a dry product to save transport costs and extend storage life [14].

The equilibrium relative humidity (RH) and moisture content (MC) of goods are critical considerations when constructing postharvest processing and storage systems [13]. Dried food products may undergo microbial growth, discoloration, phenolic and antioxidant degradation, and phytochemical changes due to inappropriate storage conditions [15,16]. Thus, adhering to appropriate storage temperatures, RH, and MC is critical to ensure the highest possible product quality and safety [17].

UPS is distributed as a dried product after harvest and faces many storage-related issues despite being rich in natural physiologically active substances and having excellent health functional properties. However, the effect of RH and storage times on the microorganisms, physicochemical characteristics, and phenolic and antioxidant activities in dried UPS have not been intensively studied. Therefore, this study aimed to investigate the growth of microorganisms, caking index, fucoxanthin, ascorbic acid (AA), total phenolic contents, antioxidant activities, and fatty acid composition in dried UPS powder at different RH during storage.

## 2. Materials and Methods

### 2.1. Materials and Chemicals

The sporophyll (Miyeokgui) of cultured Korean U. *pinnatifida* (Miyeok) was acquired in a dried state from HarimBio Co., Lted (Wando, Republic of Korea). The grinding process was accomplished using a specially designed grinding system [18]. Grinding was performed under a vacuum to minimize exposure to oxygen throughout the process. Before grinding, the vacuum level was established [19]. All valves were sealed off to preserve the set conditions once the absolute pressure reached 2.67 kPa and the percentage vacuum reached 97.4%. It was opened after grinding, and nitrogen gas was flushed at a rate of 5.48 L/min until the pressure in the chamber was seen on the vacuum gauge to have returned to the normal ambient pressure.

Different salts to regulate the RH levels of UPS and reagents of analytical grade in the form of anhydrous sodium sulfate, sodium hydroxide, potassium hydroxide, potassium iodine, sodium thiosulfate, Folin–Ciocalteu’s reagent, 2,2-diphenyl-1-picrylhydrazyl (DPPH), 2,2′-azino-bis (3-ethylbenzthiazoline-6-sulfornic acid) (ABTS), L-AA, sodium chloride (NaCl), sodium carbonate, gallic acid, ethanol, t-butylated hydroxyanisole (BHA), 2,4,6-tripyridyl-s-triazine (TPTZ), hydrochloric acid, iron trichloride, acetonitrile, n-hexane, isopropanol, fucoxanthin, and α-, β-, γ-, and δ- tocotrienols were purchased from the company Sigma-Aldrich (St. Louis, MO, USA).

### 2.2. Experimental Procedure

Eight different salts were chosen, including LiCl, KCH_3_CO_2_, MgCl_2_, K_2_CO_3_, Mg(NO_3_)_2_, KI, (NH_4_)_2_SO_4_, and KNO_3_ (Daejung Chemical & Metals Co., Ltd, Seoul, Korea), to have 11%, 23%, 33%, 43%, 53%, 69%, 81%, and 93% RH, respectively. The experimental setup comprised 64 desiccators (8 × 8 batches, each containing 25 g UPS powder). The 25 g samples were uniformly dispersed on five sterile aluminum Petri plates and transferred to disinfected and sterile desiccators equipped with stands with several 5 mm holes in diameter holding 850 mL saturated salt solution. After being relocated to a chamber with constant temperature control of 25 °C, the desiccators were hermetically sealed. The RH within the desiccators and the MC of the samples were measured at intervals until hygroscopic equilibrium was reached. After 1 week of keeping hygroscopic equilibrium, the 4 week storage period began. After 2, 3, and 4 weeks, the samples were randomly obtained from each desiccator for analysis.

### 2.3. Measurement of MC and a_w_


The MC of UPS powder was determined according to the AOAC (1995) method after dehydration at 105 ± 3 °C until it reached a constant weight. The water activity (a_w_) was evaluated using a water activity meter (AQS-2; Nagy Messsystem, Gaeufelden, Germany) at room temperature (25 °C). Before measurement, it was corrected using water (a_w_ = 1.000). The measurement was performed in triplicate.

### 2.4. Color Value

The color value profile of UPS powder affected by different RH levels was determined using a color difference meter (Minolta CR-300, Osaka, Japan). Before sample analysis, the colorimeter was calibrated with the use of a standard plate (Y = 93.7, X = 0.3030, y = 0.3200).

### 2.5. Caking Phenomenon

The caking index of UPS powder affected by different RH levels was determined using sieve shaker equipment (Porter Sand, USA). The sample was left for 4 weeks in a desiccator controlled at each RH condition for caking analysis. The sample was placed in sieve shaker equipment and vibrated in a 40-mesh Tyler sieve for 1 min to analyze the degree of caking from the weight of the powdered sample remaining in the sieve [20].

### 2.6. Microbiological Analysis

UPS was examined for the presence of aerobic bacteria, yeast, and mold using a minorly modified version of the method described by Lee et al. (2021) [20,21]. The measured 5 g of UPS powder was placed in an aseptic Stomacher bag (Fisher Scientific, St. Louis, MO, USA), followed by the addition of a 10-fold sterile saline solution (0.85% NaCl) and homogenization for 120 sec using a Stomacher (Model 400; Seward Co., Worthing, UK). After the mixture was serially diluted 10-fold, a diluted sample that was 1 mL in volume was spread into a Petri film (3M Co., Minneapolis, MN, USA) of aerobic count plates or yeast and mold count plates. The Petri film was incubated for 48 h at 37 °C to detect aerobic bacteria, and 120 h at 25 °C to detect yeasts and molds. Additionally, the Petri film containing 15–300 microbial colonies was counted, and the data were shown in the form of log_10_ colony-forming units (CFU/g sample).

### 2.7. Bioactive Compound Analysis

#### 2.7.1. AA

Syringe filters with 0.45 μm were used to filter the extracts in the sample preparation step before high-performance liquid chromatography (HPLC; Ultimate 3000 Series; Dionex, Sunnyvale, CA, USA). Using HPLC with an Agilent Zorbax SB-Aq C19 column (4.6 × 250 nm, 5 μm) set to 30 °C, the filtrate was investigated. At a 1.0 mL/min flow, the mobile phase comprised 0.05 M potassium phosphate and acetonitrile (60:40). The AA was detected at 254 nm, a standard curve was used to determine the quantity of AA in UPS powder. 

#### 2.7.2. Fucoxanthin

Fucoxanthin contents were determined by HPLC [4,22]. The freeze-dried sample and ethanol were extracted at a ratio of 1:10 overnight at 25 °C in the dark. UPS powder was removed using filter paper (ADVANTEC, No. 1, diameter 110mm, Toyo Roshi Kaisha, Ltd., Tokyo, Japan). The HPLC (Ultimate 3000 Series) installed at the Central Research Facilities of Gyeongsang National University was used to quantitatively examine the extract’s fucoxanthin content. The separation column used was an Agilent Zorbax SB-Aq C19 column (4.6 × 250 nm, 5 μm), and a UV detector set to 450 nm was used. Elution of the members was carried out in isocratic mode at a flow rate of 1.0 mL/min using a mobile phase composed of acetonitrile and water in a 75:25 (*v*/*v*) ratio. The fucoxanthin content in the extracts was quantified by calculating from a standard curve of fucoxanthin.

#### 2.7.3. Tocopherols

HPLC was used to quantify the tocopherol content [23]. The filtrate was analyzed using HPLC (LC-20AD; Shimadzu, Kyoto, Japan) equipped with the LiChrospher 100 Diol column (240 × 4 mm, 5 μm; Merck, Darmstadt, Germany), with the extracts being filtered through a 0.45 μm syringe filter. The mobile phase constituted n-hexane containing 0.9% isopropanol at a 1 mL/min flow rate. The tocopherols (α-, β-, γ-, and δ-tocopherols and α-, β-, γ-, and δ-tocotrienols) were quantified at UV of 285 and 325 nm, respectively. The content of each tocopherol in the samples was determined in mg/100 g edible weight by comparing the average peak area of the standard curve of individual tocopherol compounds [24].

### 2.8. Antioxidant Activity

The methods of Blois (1958) [25] and Re et al. (1999) [26] were used to determine the DPPH and ABTS radical scavenging activities. The Benzie and Strain (1996) [27] methodology was used to conduct the FRAP assay.

### 2.9. Statistical Analysis

Results represented the mean ± standard deviation. Experimental data were analyzed by analysis of variance, and the means were compared using Duncan’s multiple range tests with significance at *p* < 0.05. All statistical analyses were performed using SAS version 9.4 (SAS Institute Inc., Cary, NC, USA).

## 3. Results and Discussion

### 3.1. MC and RH

The powder’s capacity to absorb water is determined by its composition. Numerous food powders contain water-absorbent components, such as sugars [28]. The moisture sorption data for the sample utilized are depicted in Appendix A. The isotherm curve was significantly impacted by the RH. The MC of UPS powder increased from 3.12% to 24.52% when relative humidity (RH) increased from 11% to 93% as a result of establishing equilibrium with the surrounding moisture. Estrada-Bahena et al (2022) [29] observed that the MC of green coffee beans is significantly impacted by water activity. Lee et al. (2021) [21] revealed that perilla seeds substantially influenced the isotherm curve at different RH levels ranging from 11% to 93%. The green coffee beans were stored under low water activity conditions, with the water in the beans being dehumidified and lowered. The most important are MC and temperature, affecting microbial growth in the stored products. Hyun et al. (2018) [30] indicated that it is critical to set safe storage conditions for dried foods (seaweed, kelp, and pumpkin), including storage temperature, RH, storage time, and packaging type. Microorganisms grow with increasing RH. When packaged in a sealed bag or airtight container, the packaging method maintains the original levels of total mesophilic bacteria, Escherichia coli/coliform, and yeast/mold. 

### 3.2. Color Value

Color is one of the important measures that determine the preference and quality of food products. Colors may change during storage owing to chemical or biological processes. Some chemical reactions, such as enzymatic oxidation and AA browning, can occur during storage and processing [18]. The color values, such as lightness (*L**), redness (*a**), and yellowness (*b**), of UPS powder for 4 weeks at various RH levels ranging from 11% to 93% are shown in Table 1. The *L** value decreased, whereas *a** and *b** values increased with increasing RH during storage. The 93% RH condition exhibited the lowest *L** (40.25) value and greatest *a** (−1.48) and *b** (21.95) values in UPS powders. At all RH levels, the sample stored for 2 weeks had lower *L** and *b** values and a significantly higher *a** value in the UPS powder stored for 3 and 4 weeks, respectively. Color values were not significantly different for samples maintained at the same RH of 11% to 69% for 3 and 4 weeks. These results indicated that RH significantly increased the quality degradation of UPS stored for an extended period. Therefore, color values were compatible with the appearance results after 4 weeks of RH storage (Appendix A). The color characteristics could not be determined at 81% and 93% RH for extended storage because of uncountable microbiological development. Contamination of the sample surface with mold was found at 81% and 93% RH with increasing storage time. UPS powder was discolored between 11% and 33% RH and overall browned between 81% and 93% RH. Water loss manifested in symptoms of wilting, decoloration, and a lack of crispness. Additionally, the product may become rough or mushy, unappealing to the client. Foods stored at low (11–33%) and high (81–93%) RH developed color changes, such as discoloration and browning during storage [31]. Numerous studies indicated that foods stored at high RH levels exhibited chlorophyll degradation, phenolic component oxidation, and enzymatic and nonenzymatic reactions. Results were concurrent with earlier reports on whole red pepper, pear, melon, dried jujube, samnamul, and perilla seed [29,32,33,34,35]. Modified packaging or the soup product including of antioxidants changed color substantially while storage at 50% RH for 4 weeks [36].

### 3.3. Caking Phenomenon

Caking is a negative process in which amorphous powder particles are gradually distorted until they adhere to one another, finally producing enormous agglomerates. The mechanics of caking are temperature, RH, and time-dependent [3,37,38]. As a result of the problem, the product quality degrades and the shelf life is shortened [39]. The caking index values for the sample are shown in Table 2. The caking index (%) of the UPS powder at 4 weeks did not occur in the 11% to 53% RH conditions but occurred in the 69%, 81%, and 93% RH conditions, and the caking index was 88.30%, 99.75%, and 99.98%, respectively. With a successful approach to preventing or minimizing caking of hygroscopic food powders, i.e., to add flow conditioners or anti-caking chemicals to increase their flowability and/or decrease their caking propensity [40,41,42], we found that strict moisture control, low-temperature storage, low-humidity handling, and, where available, packing of the substrate in moisture-barrier materials, all contribute to preventing or decreasing powder caking. Therefore, UPS powder can prevent or minimize caking when maintained at a relative humidity of 53% or less.

### 3.4. Microbial Safety

High MC in the environment promotes bacterial growth. Numerous disease agents and pathogens, including bacteria and viruses, are easily spread by food [43]. The microbial counts of aerobic bacteria, yeasts, and molds in UPS powder stored at different RH levels are presented in Figure 1. Aerobic bacterial concentration increased progressively in UPS powder kept at 11% to 53% RH with increased storage time, but at 69–93% RH, it increased drastically. Thus, after 4 weeks of storage, the RH 93% condition exhibited considerably higher aerobic bacterial counts (3.81 log_10_ CFU/g DW) than the 81% RH condition (5.89 log_10_ CFU/g DW) in other samples. In contrast, 81–93% RH conditions and e 11–69% RH conditions after 4 weeks indicated 1.04, 1.14, 1.55, 2.15, 2.15, and 2.72 log_10_ CFU/g DW, respectively. Aerobic bacteria significantly increased with storage time at 69% to 93% RH, but were maintained at 11% to 53% RH. Molds and yeasts were not detected in the samples stored at 11% to 53% RH. Mold and yeast colonies multiply as storage time increases. The RH of a food storage environment can alter its quality because it might change its water activity. Food will ultimately achieve moisture equilibrium with its environment, resulting in evaporation from or condensation of moisture on the item’s surface. If the water activity is critical to its safety or shelf life, it must be stored in an environment that does not significantly alter this property. The optimal water activity for most food spoilages is >0.90, whereas the minimum value is between 0.80 and 0.90. Each organism’s water activity has maximum, optimal, and minimum values for growth. Bacteria require more water activity to develop than yeasts and molds. Foods with a water activity of >0.85 are particularly perishable due to their susceptibility to spoiling and harmful bacteria. Thus, the food’s water activity dictates the microbe that may grow in it to a large part. According to similar results were reported for meat [43], perilla seeds [29], cashew nuts [44], dried shredded squid, wheat flour, sunsik, red pepper powder, roasted sesame seed [45], and samnamul [35].

### 3.5. Bioactive Compound Analysis

Recent studies have established that seaweed foods exhibit various bioactivities, including antioxidant, anticancer, anti-inflammatory, antibacterial, and antidiabetic effects [46,47,48,49,50,51,52,53]. AA, fucoxanthin, and tocopherol (such as α-, β-, and γ-tocopherol of UPS powder are stored at different RH levels Figure 3 and Figure 4). 

AA is necessary for overall health and immune cell function [44]. The body does not produce AA, and the required amount varies according to health conditions and age. This antioxidant vitamin promotes healthy cell development circulation and detoxifies cells in the entire body [47]. The sample stored at low RH for 4 weeks maintained a high AA concentration. Additionally, it held a lowered AA level under high RH conditions. It showed the highest concentration at 11% RH (20.48%) and the lowest at 93% RH (4.62%). Similar results were obtained by several authors. Jensen (1969) [54] reported that AA in seaweed powder is very sensitive as the water content increases. 

Fucoxanthin is a compound with a unique allenic link and a 5,6-monoepoxide. Fucoxanthin possesses high antioxidant effects and is a source of other carotenoids [55]. Fucoxanthin is a carotenoid found in the chloroplasts of brown algae such as *U. pinnatifida*, *Saccharina japonica*, and *Sargassum fulvellum*. Additionally, fucoxanthin has anticancer [7,48,49], antiangiogenic [50], and anti-inflammatory [51] properties. Further, fucoxanthin increased the DHA in the livers of mice. The benefits garnered considerable attention, due to the unusual method, from the food industry and nutrition researchers. These effects are unique to fucoxanthin and have not been observed with other carotenoids such as *β*-carotene or astaxanthin [52]. Fucoxanthin decreased as the RH decreased. The 11% and 23% RH was the lowest at 60.64 to 62.10 μg/g fucoxanthin, and the 81% and 93% RH was highest at 108.70 to 107.99 μg/g fucoxanthin. According to comparable investigation results, the *β*-carotene stability of nanoemulsion powder [56] and dried sweet potato chips was enhanced under high RH storage conditions [57].

A potent antioxidant, vitamin E prevents the synthesis of reactive oxygen species molecules when fat is oxidized and free radical reactions proliferate. The vitamin E group includes all compounds of tocopherol and tocotrienol with the same biological activity as α-tocopherol. Inhibiting the creation of new free radicals and neutralizing free radicals is the main function of α-tocopherol. Oxidation has been linked to several illnesses and conditions, including cancer, aging, arthritis, and cataracts [58,59,60,61]. In particular, tocopherols are vulnerable to oxygen and oxidative processes when present with catalysts, such as metal ions, UV radiation, and peroxidizing unsaturated fatty acids [48]. As a result of analyzing α-, β-, γ-, and δ-tocopherol, and tocotrienol of UPS powder, δ-tocopherol and tocotrienol were not detected. α-, β-, and γ-tocopherol showed high concentrations at high RH conditions. Jensen (1969) [54] reported a significant decrease in the chlorophyll content during wet storage of seaweeds, but carotene, tocopherol, and fucoxanthin did not significantly change. Further, Kanner et al. (1978) [62] observed that tocopherol was prone to degradation in powdered paprika held at low RH, but it remained stable at high RH. There is insufficient information to identify whether the longer induction duration at high RH in the inhibition of carotenoids and tocopherol oxidation was attributable to AA activity alone or the synergistic inhibitory action of tocopherol and AA on lipid oxidation [63].

### 3.6. Antioxidant Activity

Antioxidants decrease oxidative stress, DNA defects, malignant changes, and other types of cell damage [64]. Seaweeds have compounds that might be used as a novel dietary source for antioxidants [65,66]. As with other photosynthesizing plants, marine algae are exposed to a combination of light and oxygen, creating free radicals and additional strong antioxidant potentiation [67]. Recent research has revealed that UPS contains a range of beneficial components such as polysaccharides, polyphenols, polyunsaturated fatty acids, peptides, phytosterols, and vitamins. These components provide various health benefits including antioxidant, anticancer, anti-inflammatory, antitumor, antihypertensive, antiviral, antiobesity, and antidiabetes properties [68,69]. The antioxidant activities of UPS powder stored at different RH levels are shown in Figure 2.
Figure 2Ascorbic acid, and fucoxanthin of UPS powder during storage at different RHs: (**a**) Ascorbic acid; (**b**) Fucoxanthin. Results shown are mean ± SD (n = 3). Different letters in are express significant differences (*p* < 0.05).
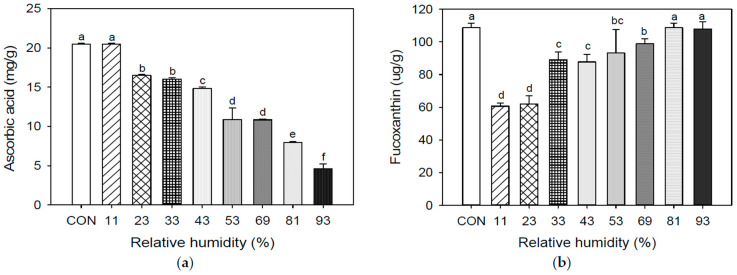

Figure 3Tocopherols of UPS powder during storage at different RHs: (**a**) α-tocopherol; (**b**) β-tocopherol; (**c**) γ-tocopherol. Results shown are mean ± SD (n = 3). Different letters in are express significant differences (*p* < 0.05).
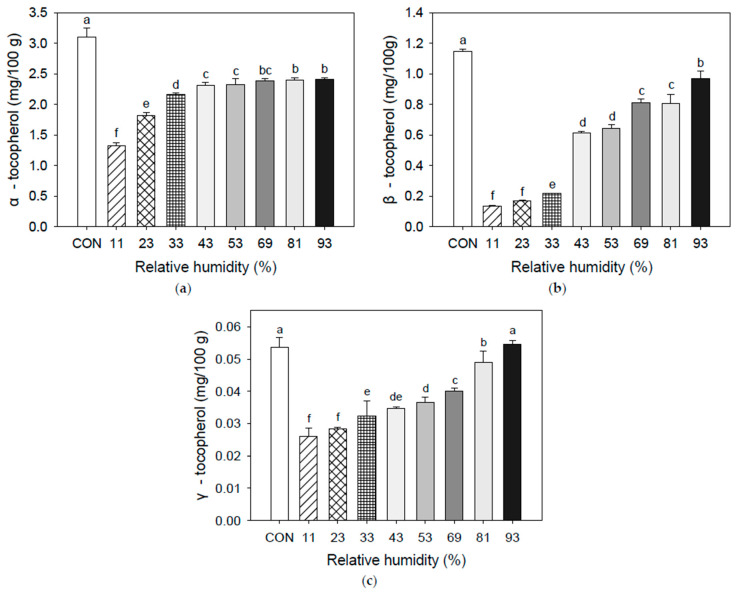


Based on their DPPH, ABTS, and FRAP, the radical scavenging capabilities of the UPS powder stored at different RH levels were examined. The DPPH antioxidant activity of UPS powder was higher than other antioxidant activities. DPPH was maintained in the samples stored at 43% to 69% RH conditions, whereas 11%, 23%, 33%, 81%, and 93% RH conditions slightly decreased. Yan et al. (1999) [70] confirmed strong DPPH activity in the organic solvent extract of edible seaweeds and identified fucoxanthin as the active component. ABTS maintained the sample stored in 69% RH condition for 4 weeks. In comparison, the 93% RH condition (3.97–2.90 g AAE/kg) had much lower activity than other samples after 2 to 4 weeks of storage. FRAP prominently decreased in the sample stored at 93% RH for four weeks. Even at 81% RH conditions, it exhibited a similar pattern to that at 93% RH. The sample stored at 69% RH conditions maintained high FRAP activity for 4 weeks. RH increased from 11.3% to 68.0%, which resulted in a drop in the antioxidant capacity and phenolic content [71]. Kim et al. (2015) [72] suggested an essential role for MC in the stability of a-tocopherol. Chen et al. (2012) [73] reported that ferric ions (Fe3^+^) dramatically increase α-tocopherol decomposition in a medium-chain triacylglycerol model system. In contrast, α- tocopherol in the presence of ferrous ions (Fe2^+^) had higher stability under the same conditions. The previous research has established a link between the seaweed extracts’ high TPC and their ability to neutralize free radicals [61,67], and in addition to fucoxanthin (carotenoid), phlorotannins (polyphenol), and tocopherols containing several antioxidants [68,69], the conversion of Fe^3+^ into this Fe^2+^ under high RH may be the cause of the instability of α-tocopherol. Recent several studies support that UPS has powerful antioxidant activity. Patra et al. (2016) [74] reported that the essential oil of UPS has potential antioxidant power in DPPH, ABTS, superoxide, and hydroxyl radical scavenging activities. Fung et al. (2013) [4] reported that antioxidant activity increased as fucoxanthin content increased. Rafiquzzaman et al. (2015) [75] reported potent antioxidant power in DPPH, ABTS, nitric oxide (NO) scavenging activities, and ferric-reducing power. It was reported to protect against oxidative DNA damage. Tong et al. (2014) [69] investigated that the UPS extract had high DPPH and hydroxyl free radical activities and was rich in phenolic compounds such as protocatechuic acid and syringic acid.
Figure 4Antioxidant activities of UPS powder during storage at different RHs: (**a**) DPPH radical scavenging activity; (**b**) ABTS radical scavenging activity; (**c**) FRAP assay. Results shown are mean ± SD (n = 3). Different letters (a–g, for the samples at different RHs, and A–G, for the same samples at different storage periods) in bars indicate significant differences (*p* < 0.05).
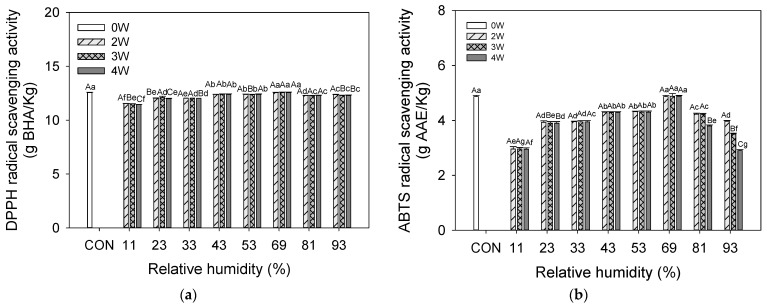

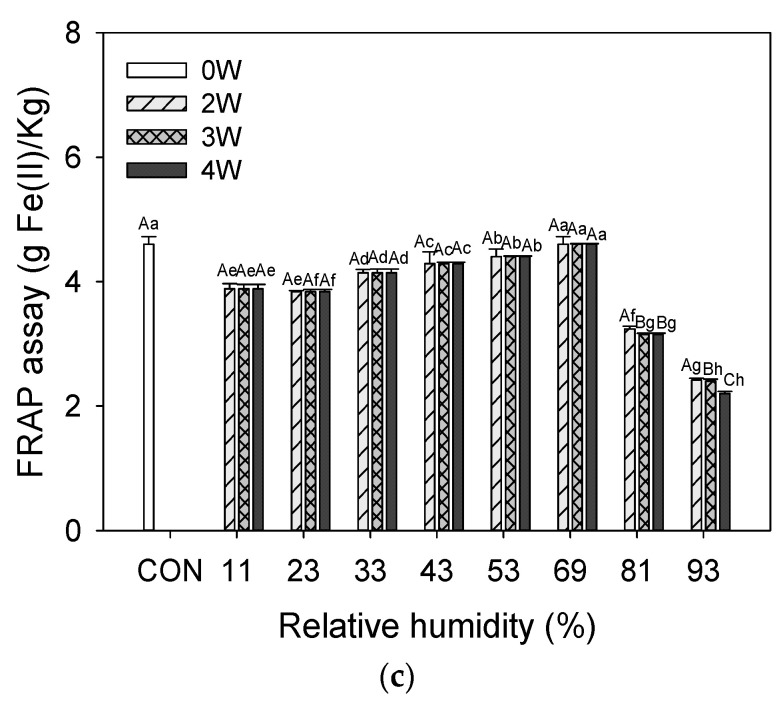


## 4. Conclusions

Dried product quality deteriorates under environmental factors like temperature and humidity throughout distribution, which may not only lead it to lose value as a product over time but also produce toxicity caused by microorganisms. In addition, in the case of dried seaweed, when the environmental conditions are humid with low-moisture food, moisture is absorbed from the surroundings into the food [21]. The RH significantly influenced the isothermal moisture absorption curve of UPS powder. The caking phenomenon began at RH 69%, and the higher the RH, the higher the caking index. During storage, aerobic bacteria, yeasts, and molds were progressively increased at RH 69–93%. Regarding the functional substance of UPSP, AA was unstable at high RH (69–93%), but fucoxanthin and tocopherol were more unstable at low RH (11–23%). For antioxidant activity, UPSP at RH 69% had higher DPPH (12.57 g BHAE/kg), ABTS (4.87 g AAE/kg), and FRAP (4.60 g Fe(II)/kg) than the other samples. This study could help with the understanding of the stability and microbial safety of UPSP during storage at different RHs.

## Figures and Tables

**Figure 1 foods-12-02342-f001:**
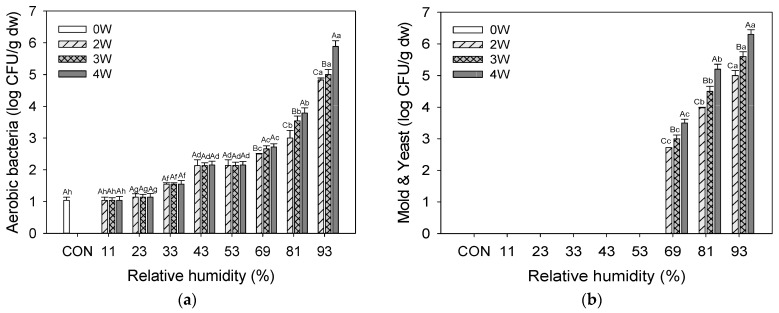
Microbial contents of UPS powder storage at different RHs. (**a**) Aerobic bacteria; (**b**) Mold and Yeast. Different letters (a–g, for the samples at different RHs, and A–G, for the same samples at different storage periods) in bars indicate significant differences (*p* < 0.05).

**Table 1 foods-12-02342-t001:** Color value of UPS powder during storage at different RHs.

Storage Time (Week)	Color Value	Relative Humidity (%)							
		11	23	33	43	53	69	81	93
2	*L*	50.13 ± 0.69 ^Aa^	48.56 ± 1.43 ^Ab^	49.15 ± 0.64 ^Ab^	48.77 ± 1.26 ^Ab^	47.19 ± 0.60 ^Ac^	44.96 ± 0.39 ^Ad^	41.27 ± 0.69 ^Ae^	40.24 ± 0.77 ^Af^
	*a*	−4.87 ± 0.07 ^Af^	−4.52 ± 0.19 ^Ae^	−4.12 ± 0.30 ^Ad^	−3.86 ± 0.04 ^Ac^	−3.88 ± 0.22 ^Ac^	−2.18 ± 0.15 ^Bb^	−2.07 ± 0.11 ^Ab^	−1.48 ± 0.21 ^Aa^
	*b*	15.47 ± 0.95 ^Af^	16.68 ± 0.62 ^Ae^	17.74 ± 0.31 ^Ad^	18.89 ± 1.05 ^Ac^	18.93 ± 0.27 ^Ac^	21.34 ± 1.15 ^Ab^	21.56 ± 0.59 ^Aab^	21.95 ± 0.84 ^Aa^
3	*L*	49.68 ± 0.07 ^Aa^	48.92 ± 0.54 ^Ab^	48.27 ± 0.85 ^Bb^	48.17 ± 0.37 ^Ac^	47.03 ± 0.14 ^Ac^	44.85 ± 0.92 ^Ad^	-	-
	*a*	−4.98 ± 0.30 ^Af^	−4.77 ± 0.26 ^Be^	−4.20 ± 0.09 ^Ad^	−3.88 ± 0.09 ^Ac^	−2.77 ± 0.26 ^Bb^	−2.17 ± 0.34 ^Ba^	-	-
	*b*	15.47 ± 1.17 ^Ad^	16.02 ± 0.37 ^Bd^	17.56 ± 0.24 ^Ac^	18.30 ± 0.17 ^Ac^	19.78 ± 0.53 ^Ab^	21.19 ± 1.06 ^Aa^	-	-
4	*L*	49.59 ± 0.99 ^Aa^	49.76 ± 0.41 ^Aa^	46.77 ± 0.42 ^Cc^	46.56 ± 0.92 ^Bc^	45.23 ± 0.35 ^Bd^	43.38 ± 0.98 ^Bd^	-	-
	*a*	−5.26 ± 0.17 ^Be^	−4.86 ± 0.12 ^Bd^	−4.76 ± 0.17 ^Bd^	−4.28 ± 0.26 ^Bc^	−2.47 ± 0.03 ^Cb^	−1.80 ± 0.09 ^Aa^	-	-
	*b*	15.06 ± 0.48 ^Ac^	15.63 ± 0.90 ^Bc^	17.54 ± 1.53 ^Ab^	17.99 ± 1.25 ^Ab^	20.03 ± 1.43 ^Aa^	19.40 ± 0.49 ^Ba^	-	-

All values are mean ± SD (n = 3). Different letters (a–f for the samples at different RHs, A–C for the same samples at different storage periods) indicate significant differences (*p* < 0.05).

**Table 2 foods-12-02342-t002:** Caking index (%) of UPS powder during storage at different RHs.

**Relative humidity (%)**	11	23	33	43	53	69	81	93
**Caking index (%)**	0	0	0	0	0	88.30 ± 0.11 ^c^	99.75 ± 0.08 ^b^	99.98 ± 0.01 ^a^

All values are mean ± SD (n = 3). Different letters indicate significant differences (*p* < 0.05).

## Data Availability

Data is contained within the article.

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
