# Peer review of "Miyeokgui* (*Undaria pinnatifida* Sporophyll) Characteristic under Different Relative Humidity: Microbial Safety, Antioxidant Activity, Ascorbic Acid, Fucoxanthin, *α*-/*β*-/*γ*-Tocopherol Contents"

_foods, 2023, doi:10.3390/foods12122342_

Round 1

Reviewer 1 Report

-        Line 2, 15, 23, 35: sporophyll should not be written in italic.

-        Line 13: “different relative humidity” percentage, index, ratio?

-        Line 16: a verb is missing: but occurred

-        Line 25: Keywords: Unrdaria

-        In the abstract the authors should conclude stating the optimum range of relative humidity percentage for the storage.

-        line 34 and 35 U from U. pinnatifida should be at italic

-        Line 35: “U. pinnatifida sporophyll (UPS) is a brown marine alga containing beneficial bioactive compounds.”: Please consider revising: U. pinnatifida sporophyll (UPS) is a part of the brown marine alga. 

-        Line 36: The Functional materials cited are present in all part of the alga or the authors or only in the sporophyll?

-        Line 38: Please correct or remove: “Fucoxanthin is a fatty acids, dietary fibers, minerals, vitamins, and phytochemicals [6].” 

-        Line 45: consider another term instead of practical, as suitable

-        Line 50: “transportion coste”: Do the authors mean transport costs? 

-        Line 82, 83: please introduce the underscript of the chemical formula

-        Line 120: (3M Co., Minneapolis, MN, USA). Which was the reference of the 3M Petri films used to detect bacteria, yeasts and molds?

-        Line 128: please reformulate: “Sample preparation before analysis was used a syringe filter with an of 0.45 μm was used to filter the extracts. Syringe filters with 0.45 μm were performed to filter the extracts of sample preparation step before” 

-        Line 138: “Fucoxanthin was determined using HPLC [4,23].” Please be more accurate, writing for example: Fucoxanthin contents were determined by HPLC.

-        Line 163: Consider reformulate: “the method was employed of Re et al. (1999) was used to assess the ABTS radical scavenging activity” 

-        Line 144: “the components were detected using an ultraviolet (UV)-visible detector set to 450 nm.” 450 nm is not a visible wavelength?

-        Line 155: “The tocopherols (α-,β-, γ-, and δ-tocopherols 155 and α-,β-, γ-, and δ-tocotrienols) were quantified at UV and visible wavelengths of 285 and 325 nm, respectively.” 185 and 325 nm are not both not UV wavelength?

-        Line 148, 158: please provide the references of all the standard used (manufacturer)

-        Line 162: “The method of Blois (1958) was used to determine the DPPH radical scavenging activity with a few minor changes. With some slight adjustments, the method was employed of Re et al. (1999) was used to assess the ABTS radical scavenging activity.” The authors have to explain what were the minor changes and adjustments introduced in the protocols 

-        Line 190: “These results corroborate previous research on rough rice, brown rice, maize kernels, corn cobs, soybeans, and red beans [34]“. Please check the reference 34 (Atalla, M.M.; Hassanein, N. M.; El-Beih, A. A.; Youssef, Y. A. G. Mycotoxin production in wheat grains by different Aspergilli in relation to different relative humidities and storage periods. Mol. Nutr. Food Res. 2003, 47, 6-10, 475 https://doi.org/10.1002/food.200390017) that is relative to wheat grains.

-        Line 210: “Contamination observed on the surface of the samples, including yeast and fungi.“ A word is lacking, contamination could be/were observed on the surface of the samples? 

-        Line 230: “but in the 69%, 81%, and 93%”, should be: but occurred in the 69%, 81%, and 93%?

-        Line 231: “With a successful approach to preventing or minimizing caking of hygroscopic food powders being to add flow conditioners or anti-caking chemicals to increase their flowability and/or decrease their caking propensity [47-49], we found that strict moisture control, low-temperature storage, low-humidity handling, and, where available, packing of the substrate in moisture-barrier materials, all contribute to preventing or decreasing powder caking.” The authors should state which is the successful approach to preventing or minimizing caking.

-        Line 269: it is “mosture absorbente” instead of “moisture absorbent “ 

-        Line 369: “cant decrease in preference (Lee et al., 2022) Therefore, UPS, a food material with a supe- “ -> “cant decrease in preference (Lee et al., 2022). Therefore, UPS, a food material with a supe-“

-        Line 369: (Lee et al., 2022)? Is this ref [22]?

-        In the discussion, it will be interesting to discuss the best way to store the UPS.

English needs serious revisions along the whole manuscript, preferably by a native english speaker.

Author Response

Response to Reviewers

Reviewer  #1

[1]

Comment: Line 2, 15, 23, 35: sporophyll should not be written in italic.

Response: As reviewer suggested, we have corrected “sporophyll” to “sporophyll” in Line 2, 16, 26, 34, and 36.

[2]

Comment: Line 13: “different relative humidity” percentage, index, ratio?

Response: As suggested, we have corrected “different relative humidity” to “different relative humidity (%)” in Line 14.

[3]

Comment: Line 16: a verb is missing: but occurred.

Response: As suggested, we have added “occur” in Line 16.

[4]

Comment: Line 25: Keywords: Unrdaria

Response: As suggested, we have corrected “Unrdaria” to “Undaria” in Line 25.

[5]

Comment: In the abstract the authors should conclude stating the optimum range of relative humidity percentage for the storage.

Response: Based on our results for caking phenomenon, mold & yeast safety, and ascorbic acid content were optimum conditions ranged from 11% to 53% relative humidity. However, optimum conditions for fucoxanthin and tocopherol contents were ranged from 81% to 93% relative humidity. Therefore, it is difficult to determine the optimum range of relative humidity to maintain all aspects such as microbial safety, functional component, and physical stability of the UPSP.

[6]

Comment: Line 34 and 35 U from U. pinnatifida should be at italic.

Response: As suggested, we have corrected (Line 33 and Line 35).

[7]

Comment: “U. pinnatifida sporophyll (UPS) is a brown marine alga containing benefical bioactive compounds.” Please consider revising: U. pinnatifida sporophyll (UPS) is a part of the brown marine alga.

Response: As suggested, we have revised “ U. pinnatifida sporophyll (UPS) is a part of the brown marine alga.” (Line 35-36)

[8]

Comment: The functional materials cited are present in all part of the alga or the authors or only in the sporophyll?

Response: The functional materials mentioned are present in brown algae.

[9]

Comment: Please correct or remove “Fucoxanthin is a fatty acids, dietary fibers, minerals, vitamins, and phytochemicals [6].

Response: As suggested, we have removed (Line 38-39).

[10]

Comment: Line 45: consider another term instead of practical, as suitable

Response: As suggested, we have corrected “suitable resource” (Line 45).

[11]

Comment: Line 50: “transportion coste” : Do the authors mean transport costs?

Response: We have revised “transportion coste” to “transport costs” (Line 49).

[12]

Comment: Line 82, 83: please introduce the underscript of the chemical formula

Response: As suggested, we have revised “LiCl, KCH3CO2, MgCl2, K2CO3, Mg(NO3)2, KI, (NH4)2SO4, and KNO3” (Line 85-86).

[13]

Comment: Line 120: (3M Co., Minneapolis, MN, USA). Which was the reference of the 3M Petri films used to detect bacteria, yeasts and molds?

Response: We have revised “After the mixture was serially diluted 10 fold, a diluted sample that was 1 mL in volume was spread into Petri film (3M Co., MN, USA) of aerobic count plates or yeast and mold count plates. (Line 122-124)

[14]

Comment: Line 128: please reformulate: “Sample preparation before analysis was used a syringe filter with an of 0.45 μm was used to filter the extracts. Syringe filters with 0.45 μm were performed to filter the extracts of sample preparation step before” 

Response: As a reviewer suggested, we have removed “Sample preparation before analysis was used a syringe filter with an of 0.45 μm was used to filter the extracts.” (Line 146-148)

[15]

Comment: Line 138: “Fucoxanthin was determined using HPLC [4, 23].” Please be more accurate, writing for example: Fucoxanthin contents were determined by HPLC.

Response: As suggested, we have corrected “Fucoxanthin was determined using HPLC.” to “Fucoxanthin contents were determined by HPLC.” (Line 141-142)

[16]

Comment: Line 163: Consider reformulate: “the method was employed of Re et al. (1999) was used to assess the ABTS radical scavenging activity”

Response: As suggested, we have changed “The methods of Blois (1958) and Re et al. (1999) was used to determine the DPPH and ABTS radical scavenging activities. (Line 165-166)

[17]

Comment: Line 96: Line 144: “the components were detected using an ultraviolet (UV)-visible detector set to 450 nm.” 450nm is not a visible wavelength?

Response: As suggested, we have revised “and using an UV detector set to 450 nm.”

 (Line 147)

[16]

Comment: Line 155: “The tocopherols (α-,β-, γ-, and δ-tocopherols 155 and α-,β-, γ-, and δ-tocotrienols) were quantified at UV and visible wavelengths of 285 and 325 nm, respectively.” 185 and 325 nm are not both not UV wavelength?

Response: As suggested, we have revised “The tocopherols (α-,β-, γ-, and δ-tocopherols 155 and α-,β-, γ-, and δ-tocotrienols) were quantified at UV of 285 and 328nm, respectively.” (Line 158-160)

[17]

Comment: Line 148, 158: please provide the references of all the standard used (manufacturer)

Response: As suggested, we have corrected “The fucoxanthin content in the extracts was quantified by calculated from standard curve of fucoxanthin.” (Line 149-151). and “The content of each tocopherol in the samples was determined in mg/100 g edible weight by comparing the average peak area of the standard curve of individual tocopherol compounds [23]. (Line 160-162)

[18]

Comment: Line 106: Writing “machine was designed with” instead of “machine designed” is suggested.

Response: We have changed sentence. “The grinding process was accomplished using a specially designed grinding system [19].” (Line 67-68)

[19]

Comment: Line 162: “The method of Blois (1958) was used to determine the DPPH radical scavenging activity with a few minor changes. With some slight adjustments, the scavenging activity.” The authors have to explain what were the minor changes and adjustments introduced in the protocols

Response: We have changed the sentence. (Line 165-166)

[20]

Comment: Line 190: “These results corroborate previous research on rough rice, brown rice, maize kernels, corn cobs, soybeans, and red beans [34]”.” Please check the reference 34 (Atalla, M.M.; Hassanein, N. M.; El-Beih, A. A.; Youssef, Y. A. G. Mycotoxin production in wheat grains by different Aspergilli in relation to different relative humidities and storage periods. Mol. Nutr. Food Res. 2003, 47, 6-10, 475 https://doi.org/10.1002/food.200390017) that is relative to wheat grains.

Response: As reviewer suggested, we have removed “Atalla, M.M.; Hassanein, N. M.; El-Beih, A. A.; Youssef, Y. A. G. Mycotoxin production in wheat grains by different Aspergilli in relation to different relative humidities and storage periods. Mol. Nutr. Food Res. 2003, 47, 6-10, 475 https://doi.org/10.1002/food.200390017” and “According to Atalla et al. (2003), the MC and temperature are critical parameters of fungl development in storage. These results corroborate previous research on rough rice, brown rice, maize kernels, corn cobs, soybeans and red beans [34] as well as wheat read [35].” (Line 191-194)

[21]

Comment: Line 210: “Contamination observed on the surface of the samples, including yeast and fungi.” A word is lacking, contamination could be/were observed on the surface of the samples?

Response: As reviewer suggested, we have corrected “Contamination of the sample surface with mold was found at 81% and 93% RH with increasing storage time.” (Line 213-214)

[22]

Comment: Line 230: “but in the 69%, 81%, and 93%”, should be: but occurred in the 69%, 81%, and 93%?

Response: As suggested, we have revised “but occurred in the 69%, 81%, and 93% RH conditions,” (Line 233-234)

[23]

Comment: Line 231: “With a successful approach to preventing or minimizing caking of hygroscopic food powders being to add flow conditioners or anti-caking chemicals to increase their flowability and/or decrease their caking propensity [47-49], we found that strict moisture control, low-temperature storage, low-humidity handling, and, where available, packing of the substrate in moisture-barrier materials, all contribute to preventing or decreasing powder caking.” The authors should state which is the successful approach to preventing or minimizing caking.

Response: As suggested, we have added “Therefore, UPS powder can prevent or minimize caking when maintained at a relative humidity of 53% or less.” (Line 240-241)

[24]

Comment: Line 269: it is “mosture absorbent” instead of “moisture absorbent”

Response: As suggested, we have corrected “moisture absorbent” (Line 273)

[25]

Comment: Line 369: “cant decrease in preference (Lee et al., 2022) Therefore, UPS, a food material with a supe-“ -> “cant decrease in preference (Lee et al., 2002). Therefore, UPS, a food material with a supe-

Response: As suggested, we have corrected “significant decrease in preference [22]. Therefore, UPS, a food material with a superior” (Line 388-389)

[26]

Comment: Line 369: (Lee et al., 2022)? Is this ref [22]?

Response: As suggested, we have revised “significant decrease in preference [22]. Therefore, UPS, a food material with a superior” (Line 388-389)

[27]

Comment: In the discussion, it will be interesting to discuss the best way to store the UPS.

Response: As suggested, we have added sentences (Line 407-413).

We hope you will consider this paper as suitable for publication in your journal.

Best regards.

*Corresponding author: Sung-Gil Choi, Professor

Division of Food Science and Technology (Institute of Agriculture and Life Sciences), Gyeongsang National University, Jinju 52828, Korea

phone : +82-10-7143-3100

Fax: +82-55-772-1909

E-mail address: sgchoi@gnu.ac.kr

Reviewer 2 Report

The manuscript titled 'Miyeokgui (Undaria pinnatifida sporophyll) characteristics under different relative humidity: microbial safety, antioxidant activity, ascorbic acid, fucoxanthin, α-/ β-/ γ-tocopherol contents' describes the various physicochemical and biological characteristics of Undaria pinnatifida sporophyll powders, locally known as Miyeokgui, stored under different relative humidity conditions. The authors conducted an analysis of the stored Miyeokgui, considering a wide range of parameters, including microbial safety, antioxidant activity, ascorbic acid, fucoxanthin, and tocopherol contents, over a storage period of four weeks. They observed significant and noteworthy variations in the stored products depending on the relative humidity conditions. The research area is intriguing, and the findings have significant implications for the storage and distribution of Miyeokgui. The manuscript is well-written; however, there are some minor spelling and grammatical errors that need to be thoroughly checked. Therefore, the following changes are required.

Top of Form

Specific comments:

1.     Introduction: The needs for the present study should be summarized at the end of the Introduction before mentioning the aim of the study.

2.     Materials and Methods: Line 64; U. pinnatifida should be Italic.

3.     Please mention the name of apparatus and instruments used for grinding the sample in vacuum condition with origin.

4.     Experimental procedure: Line 82-83; the names of the salts should be written correctly; e.g., KCH3CO2.

5.     Figure title for antioxidant activity is missing. Additionally, the superscript letters on the bar should be enlarged for easy understanding of the readers.

6.     Conclusion: Could you please suggest a suitable/ optimum condition for storage?

7.     The authors are recommended to check the following article for improving the discussion of antioxidant activity:

there are some minor spelling and grammatical errors that need to be thoroughly checked.

Author Response

Response to Reviewers

Reviewer  #2

[1]

Comment: Introduction: The needs for the present study should be summarized at the end of the Introduction before mentioning the aim of the study.

Response: As suggested, we have added a sentence (Line 56-58).

[2]

Comment: Materials and Methods: Line 64; U. pinnatifida should be Italic.

Response: As suggested, we have revised “U. pinnatifida (Miyeok)” (Line 66).

[3]

Comment: Please mention the name of apparatus and instruments used for grinding the sample in vacuum condition with origin.

Response: As suggested, we have added “The grinding process was accomplished using a specially designed grinding system [19].” (Line 67-68)

[4]

Comment: Experimental procedure: Line 82-83; the names of the salts should be written correctly; e.g., KCH3CO2

Response: As suggested, we have revised “LiCl, KCH3CO2, MgCl2, K2CO3, Mg(NO3)2, KI, (NH4)2SO4, and KNO3” (Line 85-86).

[5]

Comment: Figure title for antioxidant activity is missing. Additionally, the superscript letters on the bar should be enlarged for easy understanding of the readers.

Response: As suggested, we have added “Figure 4. Antioxidant activities of UPS powder during storage at different RHs. Different letters (a-g, for the samples at different RHs, and A-G, for the same samples at different storage periods) in bars indicate significant differences (p<0.05). (Line 377-379)” Furthermore, the superscript letters on the bar and figures have been enlarged for better viewing.

[6]

Comment: Conclusion: Could you please suggest a suitable/ optimum condition for storage?

Response: As suggested, we have added sentences (Line 407-413).

[7]

Comment: The authors are recommended to check the following article for improving the discussion of antioxidant activity:

Response: As suggested, we have added sentences (Line 344-355 and Line 364-372).

We followed the references below.

  1. Pisochi, A.M.; Pop, A. The role of antioxidants in the chemistry of oxidative stress: A review. Eur. J. Med. Chem. 2015, 97, 55-74, https://doi.org/10.1016/j.ejmech.2015.04.040
  2. Boulom, S.; Robertson, J.; Hamid, N.; Ma, Q.; Lu, J. Seasonal changes in lipid fatty acid, α-tocopherol and phytosterol contents of seaweed, Undaria pinnatifida, In the Marlborough Sounds, New Zealand. Food Chem. 2014, 161, 261-269, https://doi.org/10.1016/j.foodchem.2014.04.007.

  1. Tong, T.; Li, J.; Ko, D.O.; Kim, B.S.; Zhang, C.; Ham, K.S.; Kang, S.G. In vitro antioxidant potential and inhibitory effect of seaweed on enzymes relevant for hyperglycemia. Food Sci. Biotechnol. 2014, 23, 2037-2044, https://doi.org/10.1007/s10068-014-0277-z.

  1. Petra, J.K.; Lee, S.W.; Park, J.G.; Baek, K.H. Antioxidant and antibacterial properties of essential oil extracted from an edible seaweed Undaria pinnatifida. J. Food Biochem, 2016, 41, 1277-1279, https://doi.org/10.1111/jfbc.12278.

  1. Rafiquzzaman, S.M.; Kim, E.Y.; Kim, Y.R.; Nam, T.J.; Kong, I.S. Antioxidant activity of glycoprotein purified from Undaria pinnatifida measured by an in vitro digestion model. Int. J. Biol. Macromol. 2013, 62, 265-272, https://doi.org/10.1016/j.ijbiomac.2013.09.009.

We hope you will consider this paper as suitable for publication in your journal.

Best regards.

*Corresponding author: Sung-Gil Choi, Professor

Division of Food Science and Technology (Institute of Agriculture and Life Sciences), Gyeongsang National University, Jinju 52828, Korea

phone : +82-10-7143-3100

Fax: +82-55-772-1909

E-mail address: sgchoi@gnu.ac.kr

Round 2

Reviewer 1 Report

In relation to [7], line 35:

I did not mean that the authors should write that “U. pinnatifida sporophyll (UPS) is a part of the brown marine alga”. This sentence has to be removed. It was a comment, not a suggestion. The authors had stated that U. pinnatifida sporophyll is a brown marine alga. The fact is that U. pinnatifida is a brown marine alga and the sporophyll is a part of this alga. 

Line 33: “The body of Undaria pinnatifida is made up of a blade (lamina), midrib, sporophyll, and root-like structures [4]. A byproduct of the seaweed food business is U. pinnatifida’s reproductive organ, the sporophyll [5]. U. pinnatifida sporophyll (UPS) is a part of the brown marine alga. “

I suggest changing introducing first the alga and its beneficial bioactive compounds, and then, describing its structures and finally, introducing the UPS. For example:

Undaria pinnatifida is a brown marine alga containing beneficial bioactive compounds. The body of U. pinnatifida is made up of a blade (lamina), midrib, sporophyll, and root-like structures [4]. A byproduct of the seaweed food business is the reproductive organ, the U. pinnatifida sporophyll (UPS)[5].

U. has to be written in italic. 

The english of the manuscript is not suitable for publication. The manuscript was not checked by a native English speaker, the authors corrected the inaccuracies that I had indicated but I could not indicate all of them. Other revisions are required along the manuscript. For example:

382: “severalvariables”

385: “but also injure people owing to the toxicity brought on by microbial change.” 

387: “the color of the surface deteriorates the deterioration of the ingredients and browning,” 

402: “AA was analyzed with high content with control samples and RH 11%”

402: “Fucoxanthin was a high percent in RH 81 – 93%.” 

403: “α-, β-, and γ-tocopherol was observed that it was stable at high RH.” 

405: “At an RH of 43 to 69 percent, DPPH exhibited the high level of activity. ABTS and FRAP were observed highest activity of RH at 69%.” 

Author Response

Reviewer  #1

[1]

Comment: In relation to [7], line 35:

I did not mean that the authors should write that “U. pinnatifida sporophyll (UPS) is a part of the brown marine alga”. This sentence has to be removed. It was a comment, not a suggestion. The authors had stated that U. pinnatifida sporophyll is a brown marine alga. The fact is that U. pinnatifida is a brown marine alga and the sporophyll is a part of this alga. 

Line 33: “The body of Undaria pinnatifida is made up of a blade (lamina), midrib, sporophyll, and root-like structures [4]. A byproduct of the seaweed food business is U. pinnatifida’s reproductive organ, the sporophyll [5]. U. pinnatifida sporophyll (UPS) is a part of the brown marine alga. “

I suggest changing introducing first the alga and its beneficial bioactive compounds, and then, describing its structures and finally, introducing the UPS. For example:

Undaria pinnatifida is a brown marine alga containing beneficial bioactive compounds. The body of U. pinnatifida is made up of a blade (lamina), midrib, sporophyll, and root-like structures [4]. A byproduct of the seaweed food business is the reproductive organ, the U. pinnatifida sporophyll (UPS)[5].

  1. has to be written in italic. 

Response: As suggested, we have revised sentences (Line 33-36)

[2]

Comment: The english of the manuscript is not suitable for publication. The manuscript was not checked by a native English speaker, the authors corrected the inaccuracies that I had indicated but I could not indicate all of them. Other revisions are required along the manuscript. For example:

382: “several variables”

385: “but also injure people owing to the toxicity brought on by microbial change.” 

387: “the color of the surface deteriorates the deterioration of the ingredients and browning,” 

402: “AA was analyzed with high content with control samples and RH 11%”

402: “Fucoxanthin was a high percent in RH 81 – 93%.” 

403: “α-, β-, and γ-tocopherol was observed that it was stable at high RH.” 

405: “At an RH of 43 to 69 percent, DPPH exhibited the high level of activity. ABTS and FRAP were observed highest activity of RH at 69%.” 

Response: As suggested, we have revised sentences. (Line 373-385).
